# DOES GPT-4 HAVE GOOD INTUITION ABOUT FUNCTIONS?

## ABSTRACT

Humans inherently possess the intuition to model real-world functions such as predicting the trajectory of a ball at an intuitive level. Do Large Language Models (LLMs), trained on extensive web data comprising of human-generated knowledge, exhibit similar capabilities? This research pivots on probing such ability of LLMs (in particular, *GPT-4*) to mimic human-like intuition in comprehending various types of functions. Our evaluation reveals the potent abilities of GPT-4 not just to discern various patterns in data, but also to harness domain knowledge for function modeling at an intuitive level, all without the necessity of gradient-based learning. In circumstances where data is scarce or domain knowledge takes precedence, GPT-4 manages to exceed the performance of traditional machine learning models. Our findings underscore the remarkable potential of LLMs for data science applications while also underlining areas for improvement.

## 1 INTRODUCTION

With the swift advancements in the field of artificial intelligence, specifically in the area of large language models (LLMs), it becomes necessary to continuously evaluate their capabilities and potential (Hendrycks et al., 2020; Srivastava et al., 2022; Liang et al., 2022b). Trained extensively on an array of human-generated data, LLMs have manifested an impressive ability to emulate human thought processes, biases, and various cognitive abilities (Jakesch et al., 2023; Kosinski, 2023). This mimicry of human cognition has far-reaching implications, fostering significant breakthroughs across diverse fields including human-computer interaction (Hendrix, 1982), robotics (Liang et al., 2022a), social science (Kosinski, 2023), and healthcare (Xue et al., 2023).

This paper dives deeper into the investigation of LLM capabilities, particularly their skill to intuitively comprehend different types of functions, a capability fundamentally inherent to the human cognitive process. These functions range from relatively simple tasks such as predicting a ball's trajectory (a part of intuitive physics (Ehrhardt et al., 2018) – see Fig. 1 for a visual illustration) to more complex ones like recognizing trends in function plots. The aim of our research is to ascertain whether LLMs can replicate this form of comprehension, akin to the human mind, without needing to carry out the precise computations.

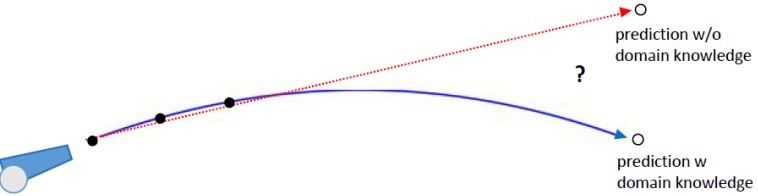

Figure 1: **Motivating example.** Consider the task of modeling the trajectory of a cannonball as a function of time. A model focusing only on function values (indicated via black points) may interpret the underlying function as a linear one (the red curve), given the nearly linear relationship depicted by the training points. However, when the domain is specified, the model can consider physical effects like air friction and drag, enabling a more accurate representation of the trajectory, which is usually parabolic, even if the data points initially suggest a linear function. With training on human-generated data, LLMs demonstrate the capability to apply domain knowledge for function modeling tasks.

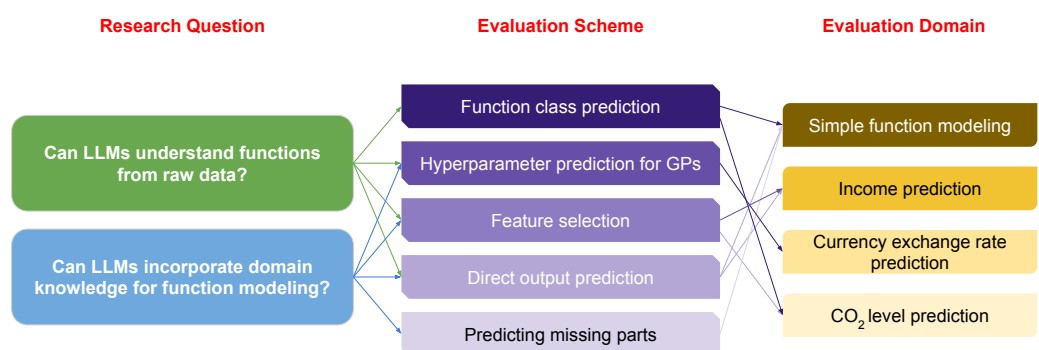

Figure 2: **Overview of our evaluation framework.** In our framework evaluation, we aim to address two primary questions utilizing various evaluation schemes. Each scheme is subsequently assessed within specified evaluation domains.

Furthermore, prior work has leveraged these function modeling capabilities in different ways, such as predicting velocities in robotics (Liang et al., 2022a), predicting the accuracy of models with different configurations for architecture search (Zheng et al., 2023), directly predicting elevation given geospatial coordinates (Roberts et al., 2023), or even direct forecasting of financial time-series data (Yu et al., 2023). In this work, we ask a more fundamental question of how capable these models are in terms of representing general functions.

Our work focuses on two key questions: (i) *can LLMs comprehend functions when presented with raw data?*, and (ii) *to what extent can they integrate and utilize domain-specific knowledge in function modeling?* Aiming to answer these questions, we undertake an exhaustive evaluation, comprising of both real-world as well as synthesized functions, while simultaneously attempting to minimize data contamination concerns. We propose a comprehensive evaluation scheme to answer these questions (summarized in Fig. 2). As static benchmarks are not ideal for these chat-based model interfaces (Bubeck et al., 2023), we consider a qualitative evaluation in order to correctly understand the capabilities of such systems. We restrict our investigation to the most powerful LLM to date i.e., **GPT-4**.

Our investigation reveals that GPT-4 is proficient in deciphering data and employing domain-specific knowledge for function modeling, using only in-context learning (Brown et al., 2020). Furthermore, in scenarios where data is scarce, GPT-4 demonstrates performance superior to the tested MLP models by capitalizing on its existing knowledge gained by looking at vast amounts of human-generated text.

While our results predominantly emphasize the promising potential of LLMs in data science applications, they also unveil potential areas for refinement and further research. In particular, the contribution of our work is threefold:

- We propose a comprehensive framework comprising different evaluation schemes in order to evaluate the function modeling capabilities of LLMs.
- Our evaluation, using synthetic and real-world data, demonstrates that LLMs are not only adept at identifying data patterns, but also highly efficient in applying domain-specific knowledge to function modeling tasks.
- We present several interesting applications leveraging the capabilities of LLMs in leveraging domain knowledge such as feature selection and kernel design for Gaussian Process models.

## 2 FUNCTION MODELING USING PRETRAINED LLMS AS BAYESIAN INFERENCE

We frame real-world function modeling tasks as performing Bayesian inference in functional space:

$$p(f|\mathcal{D}) \propto p(\mathcal{D}|f)p(f) \tag{1}$$

where $\mathcal{D}$ is the data, $p(\mathcal{D}|f)$ is the likelihood function that measures to what extent $f$ fits the data and $p(f)$ is the prior on $f$. As the posterior is dependent on both the prior as well as the likelihood, both

$p(\mathcal{D}|f)$ and $p(f)$ are important for function modeling. For example, a linear function that accurately describes the trajectory of a ball in Fig. 1 may have a high likelihood, yet a good prior from physics would identify that a quadratic function is indeed more plausible.

In this work, we are interested in the abilities of LLMs to implicitly model $p(f|\mathcal{D})$ and $p(f)$. Under this setting, the prior $p(f)$ in LLM can be seen as the posterior after seeing the massive internet-scale data $\mathcal{D}_{\text{pretrain}}$:

$$p(f) = p(f|\mathcal{D}_{\text{pretrain}}) \tag{2}$$

We evaluate such abilities of LLM by evaluating (a) $p(\mathcal{D}|f)$, where we evaluate whether LLM has good intuition about $f$ when only data $\mathcal{D}$ is given; and (b) $p(f)$, where we evaluate whether LLM can make use of any knowledge learned from the internet to form a good prior about $f$; and (c) $p(f|\mathcal{D})$, where we evaluate whether LLM can effectively combine these two kinds of understanding for function modeling.

## 3 METHODS

Accurately estimating the function modeling capabilities of LLMs is a difficult goal to realize, primarily due to the dependence on prompt tuning as well as different facets of model capabilities. Seeking to thoroughly examine multifaceted aspects of model capabilities, we propose an extensive evaluation suite. Fig. 2 offers a concise overview of our evaluation framework, which is fundamentally built around two key questions as highlighted in the figure. In this section, we will delve into both the array of tasks and the datasets that constitute our evaluation approach.

### 3.1 TASKS

**Direct output prediction.** This is the simplest kind of evaluation, and perhaps the most prevalent one in practice (Roberts et al., 2023; Yu et al., 2023), where we ask the LLM to directly predict the output (a real number) based on the input features. Successful prediction in this case is a good indicator of the model's capability in understanding and handling that function type.

**Function class prediction.** In the case of function class prediction, we ask the model to predict the function class given function values instead of predicting the function values themselves. Since this is a multi-class classification task, we provide all possible function classes considered in this case (ten for the synthetic dataset) as part of the prompt.

**Predicting missing parts.** This task focuses on identifying missing parts of a function by predicting the location and the number of values missing rather than the values themselves after specifying function constraints such as smoothness.

**Feature selection.** Domain knowledge is a powerful predictor of features that are relevant to the task alongside the data itself. Therefore, we evaluate LLM's capability in function modeling by asking it to predict the relevant features from a multivariate function. Formally, the task in this case is to select a subset of features $X' \subset X$ whose combination are the most predictive w.r.t the target $Y$: $\min_{X' \subset X} |I(X'; Y) - I(X; y)|, \quad s.t. \ |X'| = k$, where $I(\cdot, \cdot)$ denotes mutual information.

**Kernel design for GPs.** Designing the correct Gaussian Process (GP) (Williams & Rasmussen, 2006) kernel requires a complex understanding of the domain as well as the function type being modeled. Therefore, this also forms a natural test bench to evaluate LLM's function understanding capabilities by asking it to design kernels for GPs. A kernel in GP can be seen as measuring the 'similarity' between data points, and is potentially the most impactful area for human intervention when applying them to a novel problem.

### 3.2 DATASETS

#### 3.2.1 SYNTHETIC DATASETS

**Simple function modeling tasks** focuses on ten basic types of functions, including linear, quadratic, logarithmic, sine, power law, Gaussian, piecewise, step, exponential, as well as periodic linear functions. We sample the parameters for these functions randomly and evaluate the function values on a limited range of the independent variable (x in our case).

### 3.2.2 REAL-WORLD DATASETS

**Income classification.** We use the UCI Adult dataset (Becker & Kohavi, 1996). The data consists of 13 features describing an individual in the US. The task in this case is of binary classification, with the target being the person having a high income. The criteria for being high income is >50,000 USD / year. Since UCI Adult is a widely used dataset, it is likely for the LLM to have seen this during training. Therefore, in order to reduce the impact of potential test set contamination, we (a) change the names of features (e.g. 'education' → 'degree'); (b) change feature values (e.g. by adding noise to the age feature and simplifying marital status to be binary); (c) replace features with equivalent counterparts (e.g. 'hours per week' → 'hours per day'); (d) merge features (e.g. capital gain/capital loss → capital net gain), creating a transformed version of the original dataset. We use 100 samples for in-context learning.

**$CO_2$ concentration level modeling.** In this task, we would like to model the $CO_2$ concentration level $y \in \mathbb{R}$ as a function of time $t$, using the data collected in the Mauna Loa Observatory (Carbon Dioxide Research Group, 2004). We use data between 1957 and 1980 as the training set and predict the concentration level in 1990-1992. In order to reduce the impact of potential test set contamination, we (a) hide the information about the exact observatory that this data was collected from; and (b) add random Gaussian noise $\epsilon \sim \mathcal{N}(\epsilon; 1, 10^{-2})$ to the measurements.

**Currency exchange rates.** The currency exchange rate dataset comprises daily exchange rates for the top ten global currencies (CAD, EUR, JPY, GBP, CHF, AUD, HKD, NZD, KRW, and MXN), all relative to the USD for the year 2022[1]. As the cutoff time for GPT-4 is unclear due to model retraining, there is a risk of data contamination in this case. Our objective is to model the predictive relationships between a set of three currencies — CAD, JPY, and AUD (collectively represented as $\boldsymbol{y}$) — and the remaining currencies (denoted as $\boldsymbol{x}$) over a time period). This relationship is best described by the following probability density function: $p(\boldsymbol{y}|\boldsymbol{x}, t)$, where $\boldsymbol{x} \in \mathbb{R}^9$ signifies the exchange rates of the other six currencies and the three precious metals (gold, silver, and platinum), $\boldsymbol{y} \in \mathbb{R}^3$ is the three target currencies namely CAD, JPY and AUD and $t$ is the time. The targets to predict are CAD for days 50–100, JPY for days 100–150, and AUD for days 150–200, given partially observed data for these currencies on certain days and consistently observed data for all other currencies throughout the year (shown in Figure 5). To mitigate data leakage, we implemented a two-pronged approach similar to our $CO_2$ task methodology. First, we introduced random noise into the raw values to obscure direct data correlations. Second, we removed specific identifiers such as currency and metal names that could directly link the data to our training set.

## 4 UNDERSTANDING THE IMPACT OF DOMAIN KNOWLEDGE

In order to develop a thorough understanding of the function modeling capabilities of LLMs specifically when taking domain knowledge into account, we compare direct prediction performance on different function types using MSE as the metric. For the synthetic dataset, we also evaluate the function class prediction accuracy.

### 4.1 SYNTHETIC DATASETS

Fig. 3 provides an overview of the function modeling capabilities of GPT-4 with and without the specification of domain knowledge (the type of function) when using just 25 examples. We predict one value at a time in this case, by conditioning on the training points. The MSE indicates the discrepancy between the LLM prediction and the ground truth. Furthermore, in cases where domain knowledge is not specified (function type), we also evaluate the capability of GPT-4 to correctly identify the function type from the datapoints when specifying the list of 10 possible function types (function class prediction task) as highlighted in the title of the plot.

We see that specifying the function type helps GPT-4 to leverage the correct model class, such as modeling the step function correctly as a smooth function while the prediction without any domain knowledge fluctuates in a random fashion. For most function types, we see a positive impact of leveraging domain knowledge, where domain knowledge refers to the specification of the function

---

[1]The exchange rates data set can be downloaded at `http://fx.sauder.ubc.ca`

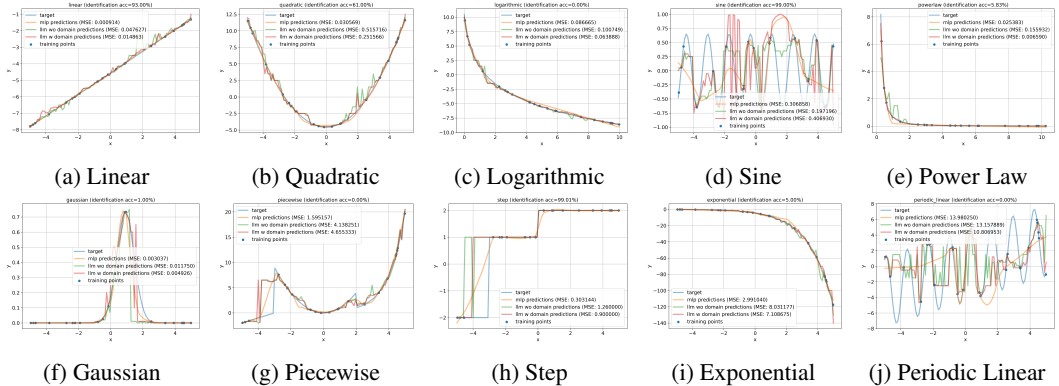

|       |       |       |       |       |
|:-----:|:-----:|:-----:|:-----:|:-----:|
| (a) Linear | (b) Quadratic | (c) Logarithmic | (d) Sine | (e) Power Law |
| (f) Gaussian | (g) Piecewise | (h) Step | (i) Exponential | (j) Periodic Linear |

Figure 3: Basic evaluations of function modeling using 25 training points, where we compare LLM performance with and without domain knowledge against a 4-layer MLP with 64 hidden units. The MSE indicates direct prediction performance, while the identification accuracy in the title of the plot indicates the function class prediction performance. See Appendix A for more results.

type. Interestingly, GPT-4 mostly fails to identify the function type except for the simplest cases i.e., linear function, step function, and the sine function.

Appendix A presents the results on the synthetic modeling task when considering a smaller number of training points as well as the addition of Gaussian noise to the data points, where we fix Gaussian noise $\sigma$ to be 0.5. It is worth emphasizing that prompting the model in a different way can change the results significantly.

## 4.2 REAL-WORLD DATASETS

In the results below, we use 'LLM' to represent the case where only raw data is provided (where we convert all features to normalized floating-point numbers), and 'LLM w/ domain' to represent the case where the context and the meaning of features are provided.

**Income classification**. Table 1 summarizes our direct prediction results for income classification. The table indicates a striking difference in performance with and without the specification of domain knowledge, indicating that GPT-4 is capable of leveraging domain knowledge effectively. Interestingly, GPT-4 with domain knowledge matches the performance of an MLP trained on two orders of magnitude more data. Due to the limited context length of GPT-4, reporting numbers beyond $n = 10^2$ was difficult, and hence, left out.

To better understand the impact of domain knowledge as well as how GPT-4 makes use of this information, we ask GPT-4 about the underlying prediction rules it uses in the two cases separately. Below are the rules GPT-4 uses with the specification of domain knowledge.

- *Degree*. Higher education (Masters, Doctorate, Bachelors) tends to have higher income.

- *Occupation*. Categories such as 'Exec-managerial' and 'Prof-specialty' are better paid.

- *Marital status*. Individuals with 'Married-civ-spouse' marital status are more correlated to higher income than individuals who are 'Never-married' or 'Divorced'.

- *Daily work hours*. People who work more hours per day (over 8 hours) generally earn more.

- *Capital net gain*. People with significant capital gains are more likely to have a high income.

Table 1: Comparison of direction prediction performance of LLM w/ and w/o domain knowledge on the binary income classification task, as well as a 4-layer MLP with 500 units while varying the number of training examples.

|         | LLM ($n = 10^2$) | LLM w/ domain ($n = 10^2$) | MLP ($n = 10^4$) | MLP ($n = 10^2$) |
|---------|:----------------:|:--------------------------:|:----------------:|:----------------:|
| acc (%) | $62.6 \pm 0.61$  | $79.4 \pm 0.83$            | $82.0 \pm 0.52$  | $73.9 \pm 0.45$  |

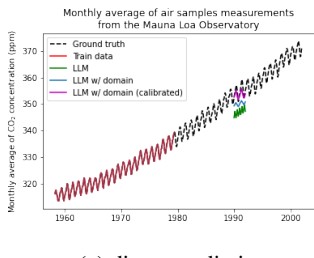
(a) direct prediction

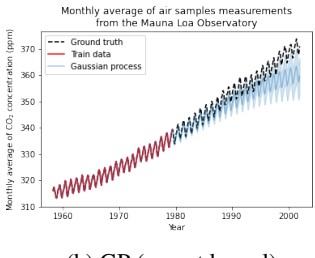
(b) GP (expert kernel)

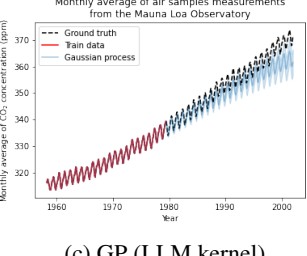
(c) GP (LLM kernel)

Figure 4: $CO_2$ concentration level modeling task. (a) Direct prediction task where GPT-4 is asked to predict the output directory. 'LLM' represents the prediction when using only raw data, while 'LLM w/ domain' represents the prediction with domain knowledge. 'LLM w/ domain (calibrated)' is asked to use all relevant rules for prediction (see Section 4.2 for description). (b-c) Predictions are made by Gaussian processes with expert-designed kernel and GPT-4 design kernel respectively.

Below are the rules that the model uses without the specification of domain knowledge:

- If feature 0 (age) and feature 2 (representative weight) are greater than 1, the likelihood of the sample belonging to class 1 increases.
- Higher values of feature 2 (representative weight) often correspond to Class 1, while values around 0 seem more correlated with Class 0;
- If feature 8 (gender) is 0 and feature 0 (age) is negative, the sample is more likely to be Class 1.

Comparing the two sets of rules, we discover that when domain knowledge is available, GPT-4 tends to make use of robust features (e.g. degree, occupation, capital net gain) that align well with common sense, whereas when there is no domain knowledge, it relies more on spurious features (e.g. representative weight, gender) that may only be predictive in the 100 training samples. This difference highlights that GPT-4 can effectively utilize domain knowledge to improve its function modeling capabilities. It is important to note that GPT-4 can misrepresent the actual decision rule. Therefore, this evaluation just validates that GPT-4 has knowledge about the domain, rather than validating what GPT-4 is doing under the hood (Huang et al., 2023).

**$CO_2$ level modelling**. Table 2 summarizes our direct prediction results on the $CO_2$ dataset (see Fig. 4a for a visual illustration). Consistent with prior results, domain knowledge plays a critical role in achieving low MSE. Interestingly, predictions made by the LLM, though being intuitive, can even outperform an expert-designed Gaussian process (GP) model (Williams & Rasmussen, 2006). Despite the LLM not being able to carry out the precise computations, this highlights the effectiveness of utilizing domain knowledge for improved function modeling.

To gain better insights into the LLM reasoning process, we ask GPT-4 to output the rules that it used for the prediction (this is akin to chain-of-thought prompting (Wei et al., 2022)). Following are the rules GPT-4 mentioned to be using:

1. *Trend*: The data exhibits a clear upward $CO_2$ concentration trend from 1958 to 1975. This aligns with established knowledge that human activities like fossil fuel burning, deforestation, and industry contribute to rising $CO_2$ levels.

2. *Seasonality*: Beyond the trend, there's a seasonal pattern. $CO_2$ concentrations peak during early Northern Hemisphere spring due to reduced plant growth, and they reach a minimum during early fall when plant growth peaks. This reflects the cycle of plant photosynthesis and respiration.

Table 2: $CO_2$ concentration modeling: Direct prediction performance of GPT-4 with and without domain knowledge as well as Gaussian process prediction with expert designed kernel and GPT-4 designed kernel.

|  | GPT-4 | GPT-4 w/ domain | GP (expert kernel) | GP (GPT-4 w/ domain kernel) |
|---|---|---|---|---|
| MSE | $66.12 \pm 4.75$ | $3.35 \pm 0.67$ | $13.78 \pm 0.01$ | $12.09 \pm 0.02$ |

Table 3: Income classification: Comparing the features selected by GPT-4 and that by state-of-the-art (Yamada et al., 2020). Accuracy (test accuracy) is measured by an MLP trained using $10^4$ samples.

|  | GPT-4 w/ domain ($n = 10^2$) | Yamada et al. (2020) ($n = 10^4$) | Yamada et al. (2020) ($n = 10^3$) |
|---|---|---|---|
| top 5 features | {degree, occupation, martial status, hours per day, capital net gain} | {degree, age, martial status, hours per day, capital net gain} | {education, education-num, gender, capital net gain, native-country} |
| accuracy (%) | 82.63 ±0.40 | 82.95 ±0.38 | 80.01 ±0.52 |

3. *Increasing Rate of Change*: Notably, the rate of $CO_2$ increase is accelerating. This corresponds with the 20th-century surge in industrial activity and energy use, particularly the burning of fossil fuels, which contributes significantly to the intensified $CO_2$ buildup rate in the atmosphere.

GPT-4 can effectively identify the main patterns in the data (Rule 1 and Rule 2) based not only on the data but also on the domain knowledge. Notably, while the data itself does not exhibit a significantly increasing changing rate (i.e. Rule 3), GPT-4 can still identify it using prior knowledge. When performing prediction, GPT-4 mainly makes use of Rule 1 (trend) and Rule 2 (seasonality) above (visualized in Fig. 4a) which slightly underestimates the true concentration level. However, explicitly hinting GPT-4 to use all three rules to account for the increasing rate of change in $CO_2$ emissions leads to highly accurate predictions (represented by 'LLM w/ domain (calibrated)' in Fig. 4a). See Appendix C for the prompts used. This is a potential reason why GPT-4 can outperform GPs in this case as the training data itself does not explicitly dictate a strongly increasing rate of change.

## 5 Extensions of Domain Knowledge for Novel Applications

After establishing the capabilities of GPT-4 in leveraging domain knowledge, we focus on extensions to non-trivial applications such as feature selection and kernel design for GPs.

### 5.1 Feature Selection

For effective feature selection, one must simultaneously understand both (a) the individual contribution of each feature, as well as (b) how features interact with each other to give rise to the underlying function (e.g., synergy effect, redundancy, etc.). Therefore, this forms a good benchmark to evaluate the capabilities of LLMs in leveraging both data and domain knowledge for function modeling.

Table 3 summarizes the results for the income dataset, where we compare the features selected by GPT-4 with those selected by a state-of-the-art feature selection method (Yamada et al., 2020). Being a data-driven method, it does not take domain knowledge into account, relying only on $n = 10^4$ samples which is two orders of magnitude larger than the dataset size used by GPT-4. Leveraging its domain knowledge, GPT-4 can effectively select a good subset of features that are comparable in terms of performance to Yamada et al. (2020), while being two orders of magnitude more data efficient. GPT-4 differs from Yamada et al. (2020) only by a single feature, where it picks occupation instead of age as a more reliable measure of income.

### 5.2 Kernel design for GPs

The design of the kernel in a GP reflects assumptions about the underlying function being modeled (Williams & Rasmussen, 2006). Therefore, this forms a natural test bench to understand the function modeling capabilities of LLMs based on domain knowledge. We evaluate this on $CO_2$ and currency exchange datasets, where we ask GPT-4 to suggest the detailed design of a GP with only raw data, or a combination of both raw data and domain knowledge. Specifically, we are interested in evaluating whether GPT-4 can come up with better kernels by incorporating domain knowledge, and hence, replace the cumbersome manual task of identifying appropriate functional assumptions for any given application.

Table 4: Comparing MSE between different kernel designs, averaged over the three target currencies CAD, JPY, and AUD. The standard deviation is calculated over 5 independent runs.

|  | GP (Human expert) | GP (GPT-4 w/o) | GP (GPT-4 w/ domain) |
|---|---|---|---|
| MSE | $0.0627 \pm 0.0000$ | $0.8593 \pm 0.0000$ | $0.0644 \pm 0.0021$ |

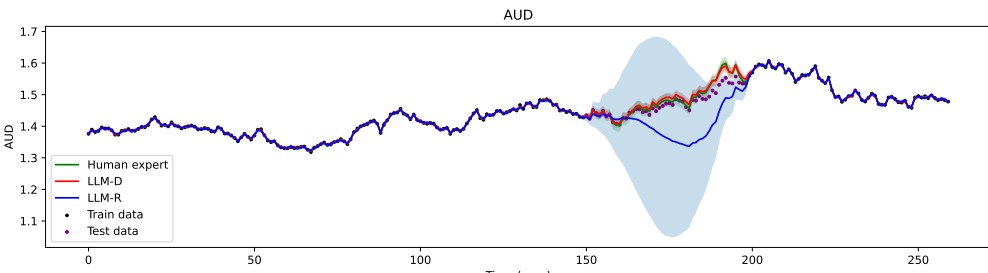

Figure 5: Kernel design comparison for GP on currency data where we compare the performance of the human expert designed kernel against the kernel designed by GPT-4 without and with domain knowledge.

**Currency exchange rates.** For this dataset, we first provide downsampled[2] raw value of currencies to GPT-4. GPT-4 without domain knowledge recommends a straightforward RBF kernel coupled with a white noise kernel, which is often considered a default setting for a GP. In this case, the input vector $\boldsymbol{x}$ predicts each of the three target currencies independently, spanning both the time and currency dimensions. Given the inherent differences between time and currency exchange rates, they should be represented differently in the underlying functions. However, GPT-4 without domain knowledge struggles to consider these properties solely from raw data, leading to a sub-optimal kernel design.

After the specification of domain knowledge, GPT-4 instead suggests the following kernel:

$$k([t, \boldsymbol{x}], [t', \boldsymbol{x'}]) = k_{llm}(t, t') + k_{llm}(\boldsymbol{x}, \boldsymbol{x'}) = k(t, t') + \sum_{j=1} k_j(\boldsymbol{x}, \boldsymbol{x'}) \tag{3}$$

with the specific design for $k_j$ shown in Table 5. Let us first look at this design qualitatively. First, it uses different kernels for the time dimension $t$ and the currencies dimensions $\boldsymbol{x}$. This design reflects that LLM understands that time and currency exchange rates are intrinsically very different objects. Therefore, they should be modeled differently in the underlying function[3]. Second, GPT-4 also elucidates general characteristics of the currency market in the kernel design, such as its volatile nature and propensity for short-term fluctuations. Interestingly, when GPT-4 with domain knowledge is asked to validate its design on the downsampled data again, it suggests discarding the periodic kernel due to the absence of clear periodicity in the data. At the same time, it also suggests a shorter hyperparameter for the length-scale, based on the potential increase in volatility towards the end of the dataset. This indicates that GPT-4 can take into account both domain knowledge and data at hand for effective GP kernel design.

Table 4 provides a quantitative comparison between the kernel designed by GPT-4 w/ domain with that by machine learning experts (Requeima et al., 2019)[4]. The table shows that the performance of the kernel designed by GPT-4 with domain knowledge is highly comparable to that of the expert design, while also being superior to the default RBF choice. Being able to design a suitable kernel suggests that GPT-4 has a good understanding of the underlying function conditioned on the provided domain knowledge.

---

[2]Downsampling is required here due to the limitation of LLMs on the maximum token number to deal with.

[3]More specifically, GPT-4 suggests this design after we clarify that we have both time and currency dimensions in the features when explaining the meaning of each feature. This results is stable across 5 trials.

[4]Note that the numbers reported for the machine learning experts are taken from recent research papers, as this forms a reliable proxy in our case.

Table 5: A comparison between human expert design, GPT-4 design given raw data (GPT-4 w/o domain), and additional domain knowledge (GPT-4 w/ domain)

| | $k([t, \boldsymbol{x}], [t', \boldsymbol{x'}])$ |
|---|---|
| Human expert | $k_{RBF}(t, t') + k_{Linear}(\boldsymbol{x}, \boldsymbol{x'}) + k_{RQ}(\boldsymbol{x}, \boldsymbol{x'})$ |
| GPT-4 w/o domain | $k_{RBF}([t, \boldsymbol{x}], [t', \boldsymbol{x'}]) + k_{Whitenoise}(\boldsymbol{x}, \boldsymbol{x'})$ |
| GPT-4 w/ domain | $k_{RBF}(t, t') + k_{Linear}(\boldsymbol{x}, \boldsymbol{x'}) + k_{RBF}(\boldsymbol{x}, \boldsymbol{x'}) + k_{Whitenoise}(\boldsymbol{x}, \boldsymbol{x'})$ |

Table 6: $CO_2$ task: Kernel designed by GPT-4 with domain knowledge and how it relates to data patterns/domain knowledge.

| | kernel type | data patterns/domain knowledge |
|---|---|---|
| $k_1$ | RBF (with large scale) | long-term trend of $CO_2$ emission |
| $k_2$ | RBF·ExpSineSquared | the yearly cyclical pattern |
| $k_3$ | Rational Quadratic | short-term fluctuations |
| $k_4$ | White Noise | observation noise and unmodeled factor |

**CO$_2$ concentration level modeling.** For the $CO_2$ dataset, GPT-4 with domain knowledge proposed the following kernel:

$$k_{llm}(t, t') = \sum_{l=1}^{4} k_l(t, t') \tag{4}$$

with each of the kernel designs $k_1, k_2, k_3, k_4$ shown in Table 6. These kernel designs are either motivated by the data patterns observed or the domain knowledge. This kernel design yields performance comparable to that of an expert design (Williams & Rasmussen, 2006), as shown in Table 2. Further details regarding the human expert design are presented in Appendix D.

## 6 RELATED WORK

Our work centers around evaluating the function modeling capability of LLMs. Liang et al. (2022a) used LLMs to convert context-dependent terms such as 'more' or 'less' to exact velocity values. This can be seen as modeling the function between the terms and the velocity. Zheng et al. (2023) used GPT-4 to predict the accuracy of models with different configurations as a more efficient architecture search scheme, which can be seen as modeling the function behind hyperparameter configurations and accuracy. GPT4GEO (Roberts et al., 2023) attempted to leverage GPT-4 to directly predict elevation given geospatial coordinates, thereby also modeling a real-world function. Yu et al. (2023) evaluated the capabilities of LLMs to directly forecast financial time series, similar to our currency exchange modeling task. Similarly, Gruver et al. (2023) used LLMs to directly forecast time-series values while demonstrating their capability to model distributions. Qin et al. (2023) conducted a comprehensive empirical assessment of GPT-4's capabilities across a spectrum of arithmetic reasoning tasks. Bubeck et al. (2023) also presented an example of function modeling tasks by evaluating the capabilities of GPT-4 in solving math riddles. Different from these works that focus on a particular domain, our work is dedicated to answering a more fundamental question, i.e., *what are the general capabilities and limitations of LLMs in comprehending functions, specifically when domain knowledge is provided?* We answer this from a Bayesian perspective, where we separately evaluate LLM's abilities to model the likelihood $p(\mathcal{D}|f)$ (i.e. intuition about raw data) and the prior $p(f)$ (i.e. intuition from domain knowledge).

## 7 CONCLUSION

Our paper presents a comprehensive evaluation of the function modeling capabilities of GPT-4 by covering several different tasks as well as several datasets (including both real-world and synthetic datasets). Primarily, our analysis revealed the potent capabilities of GPT-4 in leveraging domain knowledge in order to improve its function modeling capabilities.

**Limitations**. Our analysis focused exclusively on GPT-4, currently regarded as the most capable LLM. Furthermore, despite our best efforts to tune the prompts for the model, these results only

indicate a lower bound on performance as performance can be improved by better prompt tuning. Finally, despite our efforts to avoid data leakage through data normalization and feature engineering, we recognize that potential data leakage may persist, pointing to the need for more rigorous leakage prevention techniques in the future.

**Future work**. Our analysis was focused on GPT-4. However, given the recent surge of open-source LLMs, it is desirable to gain a better understanding of their function modeling capabilities. Simultaneously, other domains can also be evaluated such as medicine and law. Finally, taking the vision capabilities of GPT-4V will be more interesting as humans primarily develop intuition via function plots.

## REPRODUCIBILITY

To aid reproducibility, we provide details about the model and dataset in the main paper. Importantly, we provide prompts used for experiments in Appendix.

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

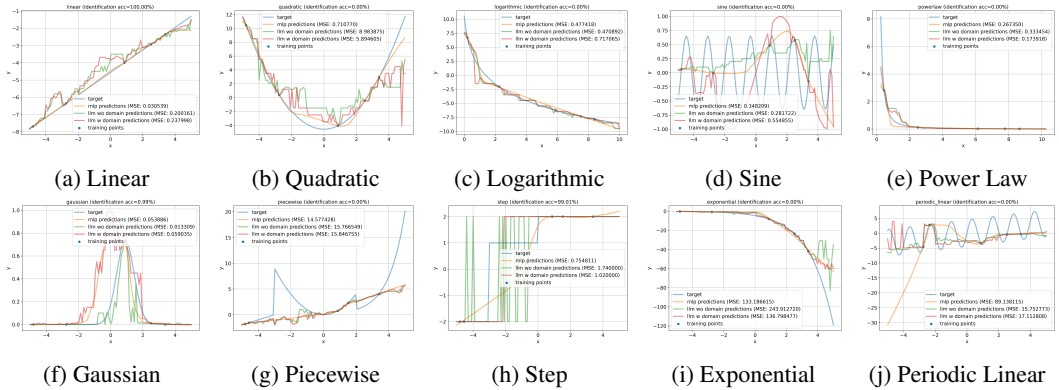

(a) Linear    (b) Quadratic    (c) Logarithmic    (d) Sine    (e) Power Law

(f) Gaussian    (g) Piecewise    (h) Step    (i) Exponential    (j) Periodic Linear

Figure 6: Basic evaluations of function modeling using 5 training points w/o any noise.

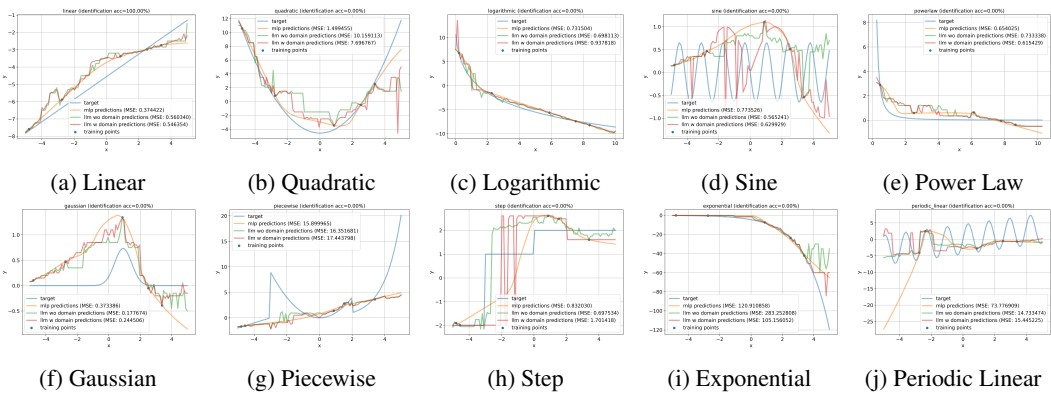

(a) Linear    (b) Quadratic    (c) Logarithmic    (d) Sine    (e) Power Law

(f) Gaussian    (g) Piecewise    (h) Step    (i) Exponential    (j) Periodic Linear

Figure 7: Basic evaluations of function modeling using 5 training points after adding Gaussian noise using standard deviation of 0.5.

## A   SYNTHETIC FUNCTION MODELING

We visualize the results for synthetic function modeling using just 5 training examples in Fig. 6. The results after additive Gaussian noise with $\sigma = 0.5$ are presented in Fig. 7.

The results with 10 training examples without and with additive Gaussian noise are presented in Fig. 8 and Fig. 9 respectively.

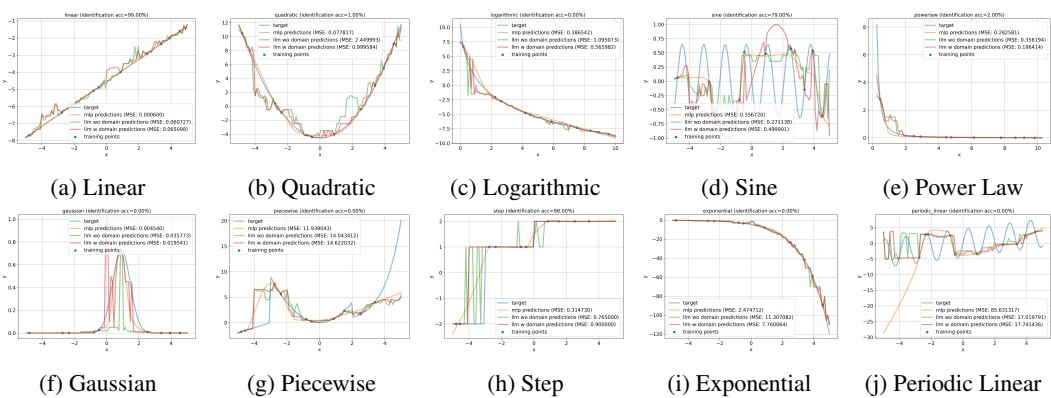

(a) Linear    (b) Quadratic    (c) Logarithmic    (d) Sine    (e) Power Law

(f) Gaussian    (g) Piecewise    (h) Step    (i) Exponential    (j) Periodic Linear

Figure 8: Basic evaluations of function modeling using 10 training points w/o any noise.

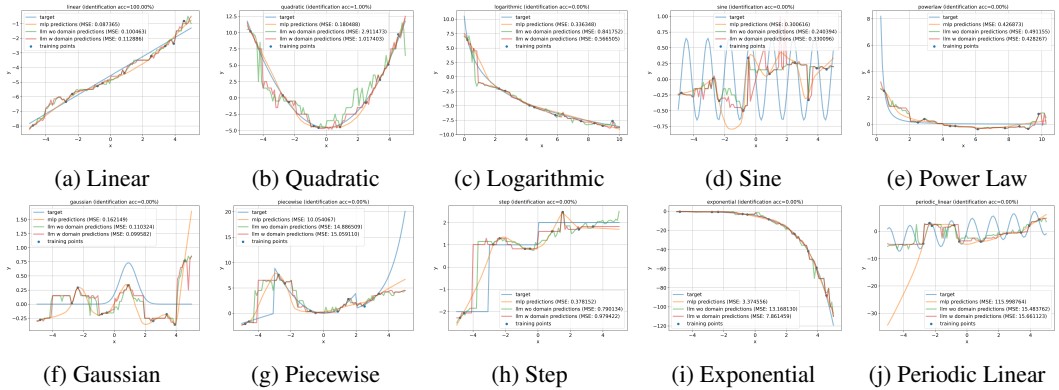

Figure 9: Basic evaluations of function modeling using 10 training points after adding Gaussian noise using standard deviation of 0.5.

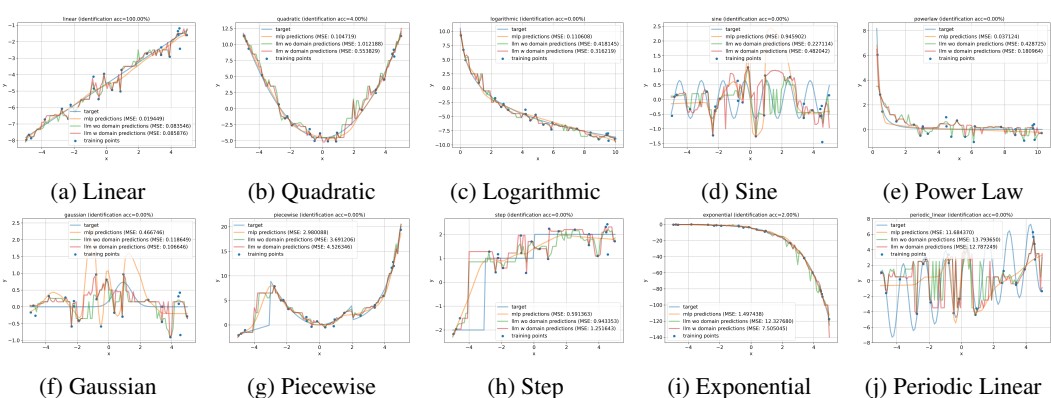

Figure 10: Basic evaluations of function modeling using 25 training points after adding Gaussian noise using standard deviation of 0.5.

Finally, the results with 25 training examples and additive Gaussian noise are presented in Fig. 10 (see Fig. 3 in the main text for the no noise counterpart).

## B    OVERVIEW OF DATA AND TASKS

Table 7: A summary of the real-world task considered in this work.

|  | $CO_2$ | Income | Currencies |
|---|---|---|---|
| Input | Univariate | Multivariate | Multivariate |
| Output | Univariate | Univariate | Multivariate |
| Domain | climate science | socio-economics | finance |
| ML model | GP | NN | GP |
| Eval. method | {direct prediction, kernel design} | {direct prediction, feature selection} | {kernel design} |

## C    PROMPTS

### C.1    SYNTHETIC DATA

**Task and setup description**
In the following task, I am going to give you some data x, y, and ask you to tell the possible relationship between x and y.

During this process, you need to directly output your answer without resorting to external softwares or libraries. Your answer needs not to be very accurate; however, providing code example or general data analysis guidance will not be acceptable.

Do you understand the setup? You are welcome to ask me any question.

**Pattern inference - 1**
Below is the data:

x= ...
y= ...

Can you tell me what is the possible relationship between x and y? A rough/intuitive estimate is enough.

**Pattern inference - 2**
Can you suggest two or three important characteristics of the function behind this data?

**Domain inference**
What do you think this dataset come from (provided that it is collected in real-world settings)?

## C.2 INCOME

**Task and setup description**
I am going to let you judge if an individual in the US has high income, using some features describing that individual as input. The features are:

['age', 'workclass', 'rep. weight', 'degree', 'marital status', 'occupation', 'relationship', 'ethnicity', 'gender', 'capital net gain', 'hours per day', 'native-country']

Below are some examples:
id:1, features: [38.8909, ' State-gov', 77516, ' Bachelors', 'not married', ' Adm-clerical', ' Not-in-family', ' White', ' Male', 2174, 8.0, ' United-States'] , high income? False
id:2, ...

Do you understand the setup? You are welcome to ask me questions if there is any uncertainty.

**In-context learning**
Below are some additional samples you can use to learn. When you learn from these samples, please combine what you learn from the data and any domain knowledge you have about this task. The samples are:

id:1, features: [38.8909, ' State-gov', 77516, ' Bachelors', 'not married', ' Adm-clerical', ' Not-in-family', ' White', ' Male', 2174, 8.0, ' United-States'] , high income? False
id:2, ...

Let me know when you already finish learning from these samples. At the same time, feel free to ask for more samples for learning if necessary.

**Direct prediction**
Now let us proceed to make some predictions. During this process, you are required to directly answer yes or no for each sample. Your answer needs not to be very accurate; an intuitive estimate is enough. However, providing code or general data analysis guidance will not be acceptable.

Below are the 25 samples I would like you to predict:

id:10000, features: [22.4297, ' Private', 176486, ' Some-college', 'not married', ' Other-service', ' Other-relative', ' White', ' Female', 0, 5.0, ' United-States'],
id: 10001, ...

Can you make a reasonable guess on whether these people are of high income or not, combining what you learned from the data above and any domain knowledge?

**Feature importance ranking**
If you were to rank the importance of each feature using a score between 0 to 1, what would you do? Again, please jointly consider the data provided as well as prior knowledge.

**Feature selection**
Now instead of giving a score for each feature, can you give the top 5 features whose combination are the most predictive?

**Feature selection vs feature importance**
I realise there is a difference between the top 5 features selected and the top 5 features with highest scores. Can you explain this difference?

**Asking for detailed reason for a specific feature being not selected**
(right after the last question)
Is it why 'workclass' is dropped out of the top 5? Can you explain it in more details?

## C.3   CO2

**Task and setup description**
I am going to ask you to predict the CO2 concentration level at some time based on some historical records. These historical data was collected at an observatory.

You should directly answer an estimate of the concentration level rather than resorting to external libraries (e.g. scipy). Also, showing me general guidance or code is unacceptable.

During this process, you are encouraged to take into account both the patterns in the data and the domain knowledge.

Do you understand the setup?

**Hinting to combine domain knowledge and data patterns**
Here you only need to provide a rough estimate rather than an accurate prediction comparable to a machine learning model. This is just to test your intuition about (a) the patterns in the data and (b) the domain knowledge.

Are you ready? Feel free to ask me any question before I show you the data.

**In-context learning**
Below are the data collected during the years 1958 to 1975. Here, x is the time of the data collected (where 1958.25 means 0.25 years after the year of 1958 i.e. March 1958. Others figures are similar), and y is the level of atmospheric CO2 concentration measured at the corresponding time.

x= ...
y= ...

Please learn from these data. You can truncate the dataset if you feel it too difficult to learn from the whole dataset. Let me know when you have finished learning.

**Asking about general prediction rule**
(after in-context learning)

Before doing prediction, let us discuss the patterns you observe from the data in more details. Please name two or three major patterns you have seen from the data, and ideally relates it to some domain knowledge.

**Perform direct prediction - predicting the yearly average**
Now please make predictions for the following dates respectively:

x = ...

**Perform direct prediction - detailed prediction**
The estimate you gave above corresponded to the average for the year of 1990. I still want a more detailed prediction for each of the dates mentioned. Can you do this for me, or do you think this detailed prediction too difficult? Again a rough estimate is enough.

**Hinting to account for the increasing changing rate**
As you said, there may be going up trend for the CO2 concentration level. Can you try to calibrate your prediction using this fact?

**Hinting to make more reasonable modeling regarding the changing rate**
I realise that you update the increasing rate to 1.4 from 1.2 in 1992. Would you think this change too steep when considering the whole picture?

### C.4 CURRENCY EXCHANGE RATES

**Task and setup description**

I want to test you in modeling functions. Especially, your goal is to formulate a model utilizing Gaussian Process to predict the missing part in international currencies(Alpha, Beta, and Gamma). Here are the details of the dataset. 'Exchange rates data set' consists of the daily exchange rate w.r.t. A of the top ten international currencies (B, C, D, E, F, G, H, I, J, and K) and three precious metals in the unknown year.
I want to test you in modeling functions. Especially, your goal is to formulate a model utilizing Gaussian Process to predict the missing part in international currencies(Alpha, Beta, and Gamma). You can ask questions if you want.

**Ask Kernels from domain knowledge**

To deal with such missing values, we need to devise proper kernels. To do so, I want you to incorporate your knowledge of currency w.r.t A in the unknown year. What do you know about the domain? And how can you include insight from this domain knowledge into designing kernels? Please give me code using GPyTorch.

**Ask Kernels from both domain knowledge and data**

Here are the real data, Alpha, downsampled every 5 (y[::5]). What can you capture from this real data? I want you to incorporate your domain knowledge of currency in an unknown year and the intuition from the real data for designing kernels. Please give me code using GPyTorch.

## D HUMAN EXPERT KERNEL DESIGN ON THE $CO_2$ DATASET

In this section, we showcase the kernel formulation crafted by human experts. Their design encompasses three distinct types of kernels:

1. A combination of RBF and exponential sine squared kernels tailored to capture seasonality.
2. The rational quadratic kernel employed to grasp irregularities.
3. An additional combination of RBF and white noise kernels aimed at modeling noise.

```python
from sklearn.gaussian_process.kernels import RBF
from sklearn.gaussian_process.kernels import ExpSineSquared
from sklearn.gaussian_process.kernels import RationalQuadratic
from sklearn.gaussian_process.kernels import WhiteKernel

long_term_trend_kernel = 50.0**2 * RBF(length_scale=50.0)

seasonal_kernel = (
    2.0**2
    * RBF(length_scale=100.0)
    * ExpSineSquared(length_scale=1.0, periodicity=1.0,
    periodicity_bounds="fixed")
)

irregularities_kernel = 0.5**2 * RationalQuadratic(length_scale=1.0,
    alpha=1.0)

noise_kernel = 0.1**2 * RBF(length_scale=0.1) + WhiteKernel(
    noise_level=0.1**2, noise_level_bounds=(1e-5, 1e5)
)

co2_kernel = (
    long_term_trend_kernel + seasonal_kernel + irregularities_kernel +
    noise_kernel
)
```

Listing 1: Python code for human expert kenel design on $CO_2$ data

