# OpenReview forum: "Does GPT-4 have good intuition about functions?"
_ICLR.cc/2024/Conference — Submitted to ICLR 2024_

### Official Review · Reviewer_zRGM · 2023-10-28

**Soundness:** 2 fair
**Presentation:** 3 good
**Contribution:** 2 fair
**Rating:** 5
**Confidence:** 4

**Summary:**

The paper presents an evaluation of the function modeling capabilities of GPT-4 using several different and datasets. The paper seeks to answer two key questions. (1) Can LLMs comprehend functions when presented with raw data? (2) What extent can they integrate and utilize domain-specific knowledge in function modeling? The paper considers a qualitative evaluation of the most powerful LLM to date i.e., GPT-4.

The paper claims that (1) GPT-4 is proficient in deciphering data and employing domain-specific knowledge for function modeling, using only in-context learning; (2) Where data is scarce, GPT-4 demonstrates superior performance over conventional machine learning models by leveraging its existing parameter knowledge.

**Strengths:**

1. The paper evaluates the function modeling capabilities of LLMs using synthetic and real-world data.

2. It claims that LLMs can identifying data patterns and also highly efficient in applying domain-specific knowledge to function modeling tasks.

3. It presents applications of such capabilities, e.g. feature selection and kernel design for Gaussian Process models.

**Weaknesses:**

1. It is not clear whether the performance is due to memorization in the training data, generalization, or in context-learning. Since transformers can learn functions in-context performing a form of meta-learning (see references below), it will be important to compare with this baseline.

Why Can GPT Learn In-Context? Language Models Implicitly Perform Gradient Descent as Meta-Optimizers
Damai Dai, Yutao Sun, Li Dong, Yaru Hao, Shuming Ma, Zhifang Sui, Furu Wei

What learning algorithm is in-context learning? Investigations with linear models
Ekin Akyürek, Dale Schuurmans, Jacob Andreas, Tengyu Ma, Denny Zhou

Furthermore, the data contamination prevention by perturbing some words do not seem to be adequate. The authors are encouraged to try out more rigorous techniques such as

Proving Test Set Contamination in Black Box Language Models
Yonatan Oren, Nicole Meister, Niladri Chatterji, Faisal Ladhak, Tatsunori B. Hashimoto
 https://arxiv.org/abs/2310.17623

2. The paper only focuses GPT-4. It will be important to study open source models such as LLaMA2. Comparing and contrasting LLaMA2 can provide more insights on the origin of this capability.

3. The evaluation takes a qualitative approach and yet the authors draw general conclusions.

**Questions:**

Understanding the function modeling capabilities of LLMs or transformer models in general is very interesting. However, the paper only focuses on GPT4. With only qualitative evaluation, it offers very limited insights and general takeaways.

It will be interesting to take a clean slate approach to study transformer models not training on function modeling tasks to understand how well the models can learn in context, and how performance scales with model size, data and compute. Furthermore, when adding function modeling tasks in the training data, how much better performance do we get? How much does it generalize out of distribution?

The authors are encouraged to take a quantitative approach so that one can draw convincing conclusions.

== post rebuttal ==

While I appreciate the authors' clear explanation, there are no additional results directly addressing my questions. I am happy to raise my score to "marginally below the acceptance threshold" from "reject, not good enough".

**Details Of Ethics Concerns:**

There is no ethics concern as it carries out a study on the function modeling capabilities of GPT4.

---

> ### Author Response · Authors · 2023-11-22
> **Response to reviewer zRGM [1/2]**
>
> First of all, we would like to thank the reviewer for their precious time in reviewing our manuscript, and for their very thorough and constructive feedback. We address the interesting questions and issues raised by the reviewer below:
>
> > It is not clear whether the performance is due to memorization in the training data, generalization, or in context-learning. Since transformers can learn functions in-context performing a form of meta-learning (see references below), it will be important to compare with this baseline.
>
> **Response:** Thanks for raising this very interesting question. There can be a multitude of driving factors for the performance in our case. Firstly, considering the training instances, we believe that it is very unlikely for the model to have seen the exact same points since we are sampling from a noisy distribution of the underlying function (using additive Gaussian noise, renormalization, as well as feature renaming and merging), therefore, it is likely to be model generalization. On the other hand, for domain knowledge, it should most likely be just memorization as it underlies the rules of the domain. Also, in some cases like direct prediction for income as well as CO_2 datasets, in-context learning might be playing a crucial role, which might even just be task recognition instead of learning the task itself via in-context learning as argued in prior work (Pan et al. 2023). However, investigating the exact nature of this seems beyond the scope of our current work, which instead attempts to just evaluate the capabilities of GPT-4 on general function modeling tasks. The reviewer’s suggestion might be a great avenue for future work.
>
> *Pan, J., 2023. What In-Context Learning “Learns” In-Context: Disentangling Task Recognition and Task Learning (Doctoral dissertation, Princeton University).*
>
>
> > Furthermore, the data contamination prevention by perturbing some words do not seem to be adequate.
>
> **Response:** We acknowledge the importance of developing rigorous approaches for evaluating test set contamination. However, we find that the methodology suggested in the reference is not a necessary indicator of dataset contamination in our context. In particular, the reference focuses on the impact of dataset ordering on perplexity as an indicator of contamination. While significant perplexity variations may indeed signal contamination, the absence of such variations does not guarantee the absence of contamination.  Also, we want to note that order of the sequence is particularly important in our applications as we consider time-series sequences. We highlight that our efforts to mitigate data leakage were comprehensive and may be the best one can do. Specifically, we have introduced several techniques, ranging from adding random noises, feature name obfuscation, feature merging, normalization, and beyond. These are much more comprehensive than the reference mentioned.  Finally, we would like to highlight that the reference mentioned by the reviewer came out last month i.e., after the submission deadline.
>
>
> > The paper only focuses GPT-4. It will be important to study open source models such as LLaMA2. Comparing and contrasting LLaMA2 can provide more insights on the origin of this capability.
>
> **Response:** We agree that our evaluations are particularly focused on GPT-4, and studying open-source models such as LLaMA-2 can be interesting. Instead of drawing generalizable insights about capabilities of LLMs in modeling functions, our aim was instead to identify the progress that we have made in terms of function modeling with the most advanced LLM to date. Therefore, we just focused on GPT-4. While evaluating other models is valuable and can provide information about the open-source alternatives when function modeling capabilities are desired, we consider this to be beyond the scope of our current work.
>
>
> > The evaluation takes a qualitative approach and yet the authors draw general conclusions.
>
> **Response:** We believe that we have both qualitative as well as quantitative results (see e.g. Table 1 and 3 for Income prediction, and Table 2, 4 for CO2 and Currency, respectively). Furthermore, we draw conjectures rather than broadly applicable conclusions about the capabilities of GPT-4 based on our experiments. It is impossible to know the test distribution when considering natural language interfaces, and hence reliably predict model performance.

---

> ### Author Response · Authors · 2023-11-22
> **Response to reviewer zRGM [2/2]**
>
> > It will be interesting to take a clean slate approach to study transformer models not training on function modeling tasks to understand how well the models can learn in context, and how performance scales with model size, data and compute. Furthermore, when adding function modeling tasks in the training data, how much better performance do we get? How much does it generalize out of distribution?
>
> **Response:** This is certainly interesting, but we consider this to be orthogonal to our study. Training the transformer model from scratch for this purpose was a focus on several prior work (e.g., Garg et al. 2022, Nguyen et al. 2022). Our work instead focuses on the learned prior from natural language and attempts to identify the capabilities of the most capable model to date i.e., GPT-4 in terms of function modeling tasks considering that it has seen a significant fraction of all human knowledge. The aim in this case is to identify the potential of GPT-4 in terms of assisting a data-scientist in mundane tasks. Investigating the function modeling capabilities with differences in scale, data, and compute is certainly interesting, and falls beyond the scope of our current work.
>
> *Garg, S., Tsipras, D., Liang, P.S. and Valiant, G., 2022. What can transformers learn in-context? a case study of simple function classes. Advances in Neural Information Processing Systems, 35, pp.30583-30598.*
>
> *Nguyen, T. and Grover, A., 2022. Transformer neural processes: Uncertainty-aware meta learning via sequence modeling. arXiv preprint arXiv:2207.04179.*

---

### Official Review · Reviewer_3rSb · 2023-11-02

**Soundness:** 3 good
**Presentation:** 3 good
**Contribution:** 3 good
**Rating:** 6
**Confidence:** 4

**Summary:**

This paper presents a comprehensive investigation into the capabilities of Large Language Models (LLMs), specifically GPT-4, to intuitively model functions in a manner akin to human cognition. It evaluates GPT-4's ability to discern patterns and apply domain-specific knowledge to function modeling, even with limited data availability, without relying on gradient-based learning.

The authors propose a new evaluation framework tailored to the conversational nature of LLMs and demonstrate through both synthetic and real-world data that LLMs like GPT-4 not only grasp data patterns but also effectively leverage domain knowledge.

The study further explores the practical applications of these capabilities in data science, highlighting the potential of LLMs in tasks such as feature selection and kernel design for Gaussian Process models.

The findings underscore the advanced understanding and potential uses of LLMs in various data-centric domains, showcasing their superiority in certain scenarios over traditional machine learning models.

**Strengths:**

The paper approaches a new and interesting problem, investigating the intuitive function modeling abilities of Large Language Models—a field that combines elements of artificial intelligence with processes reminiscent of human thought.

In Section 2, the authors establish a theoretical background for their study. They also develop a range of evaluation methods for different domains, indicating a thorough method for investigating GPT-4’s capabilities.

The clarity of the paper is good; it is written with conciseness and precision, making complex ideas accessible and the arguments presented easy to follow.

**Weaknesses:**

The paper focuses exclusively on GPT-4 for its experiments, which may result in conclusions that are not broadly applicable across different models. To ensure that the findings are more generalizable, it would be advantageous for future research to include comparisons with several other Large Language Models, at least in preliminary experiments.

A notable issue with the paper is the clarity of the experimental result figures; particularly, Figure 3 and Figure 7 are too blurry to interpret accurately. This could potentially hinder the readers' full comprehension of the data presented. Ensuring that these figures are presented in a clear, legible format is essential for the effective communication of the research results.

Additionally, the paper would benefit from the inclusion of specific examples for each task and dataset. Providing concrete examples can help readers better understand how the models work in practice and the nature of the tasks they are being evaluated on. This would not only clarify the methodologies used but also potentially highlight the strengths and limitations of GPT-4 in various scenarios.

**Questions:**

See the weaknesses.

---

> ### Author Response · Authors · 2023-11-22
> **Response to reviewer 3rSb**
>
> First of all, we would like to thank the reviewer for their precious time in reviewing our manuscript, and for their very thorough and constructive feedback. We address the interesting questions and issues raised by the reviewer below:
>
> > The paper focuses exclusively on GPT-4 for its experiments, which may result in conclusions that are not broadly applicable across different models. To ensure that the findings are more generalizable, it would be advantageous for future research to include comparisons with several other Large Language Models, at least in preliminary experiments.
>
> **Response:** We agree that our evaluations are particularly focused on GPT-4. Instead of drawing general conclusions about capabilities of LLMs in modeling functions, our aim was instead to identify the progress that we have made in terms of function modeling with the most advanced LLM to date. Importantly, our results revealed that even state-of-the-art LLMs like GPT-4 still have deficiency in function modeling (see e.g. the results where only raw data is presented), despite showing great potential in many function modeling tasks. Evaluation of other open-source models is certainly valuable, but we consider this to be beyond the scope of our current work.
>
>
> > A notable issue with the paper is the clarity of the experimental result figures; particularly, Figure 3 and Figure 7 are too blurry to interpret accurately. This could potentially hinder the readers' full comprehension of the data presented. Ensuring that these figures are presented in a clear, legible format is essential for the effective communication of the research results.
>
> **Response:** We are thankful to the reviewer for highlighting these issues with the figures. We will update the figures in the next iteration to make sure they are properly legible.
>
>
> > Additionally, the paper would benefit from the inclusion of specific examples for each task and dataset. Providing concrete examples can help readers better understand how the models work in practice and the nature of the tasks they are being evaluated on. This would not only clarify the methodologies used but also potentially highlight the strengths and limitations of GPT-4 in various scenarios.
>
> **Response:** Thank you for raising this. This would certainly be valuable. Although we are short on space in the main paper, we can certainly add examples in the appendix to provide better context to the readers for the next iteration of the paper.

---

### Official Review · Reviewer_rQFr · 2023-11-07

**Soundness:** 2 fair
**Presentation:** 1 poor
**Contribution:** 2 fair
**Rating:** 3
**Confidence:** 4

**Summary:**

The paper studies the ability of GPT4 to predict simple functional relationships from relatively small amounts of data. The question is motivated by psychological considerations- namely that people can display this ability in certain situations. The authors evaluate GPT4 on several tasks, including data from both synthetic and real-world functions. They also evaluate the effect of providing domain-specific knowledge about the function to GPT4.

**Strengths:**

The work is situated in a rich, and relatively under-explored problem domain. The synthetic data experiments are soundly designed. The authors give prompts for most of the experiments to aid reproducibility.

**Weaknesses:**

Conceptual Weaknesses

1. The paper is motivated by asking whether GPT4 has some knowledge of simple functions (an example is given of the trajectory of a cannonball), akin to human capabilities. In my opinion this is a solid research question, but I am concerned that many of the experiments have only a tenuous connection to this question. In particular, I am thinking of feature selection, income classification, and currency prediction tasks. These seem to be essentially classic ML tasks, without much connection to function learning per se (beyond the vacuous connection that any supervised learning problem can be cast as learning a function from inputs to outputs- but this is quite different from, e.g. the motivating example of the cannonball trajectory).  Considering that the authors motivate their work by comparing to human abilities this is a bit jarring- is there any evidence that humans are especially good at these kinds of tasks? Note that for the synthetic function prediction tasks, there actually is fairly detailed psychological data available on peoples’ performance.

2.In my opinion, the authors do not give sufficient consideration to the issue of data leakage. To their credit, they do identify this as an issue and propose specific fixes. But my concern is that there is no evaluation of whether these fixes have actually done anything to mitigate the issue.( I do realize the authors are in a bit of a bind here- on the one hand, it is important to evaluate on real-world data, but on the other, details of the GPT4’s training data are unfortunately not publicly available. ) What I would propose is something like this (using the CO2 data for example): evaluate GPT4 on (a) the raw, unmodified data, (b) the data with random noise added, and ( c ) some number of synthetic datasets constructed with a similar latent structure of a combination of a small number of Gaussian kernels. If the performance on GPT4 on (a) is nearly perfect, but the performance on (b) is significantly degraded, and close to the performance on (c ), then I would be convinced that the data leakage mitigation procedure has been at least partially successful. I’m not saying the authors have to do exactly this demonstration- but there does need to be some evaluation of the effectiveness of the leakage mitigation techniques.

3.This is a bit more subjective, but I can't help but feel that, even absent the issues described elsewhere in this review, that this paper doesn't make much of an original contribution beyond answering "how well can GPT do X" for yet another value of X.  We already know that GPT can do a lot of things quite well, and considering it was trained on data generated by humans, it is not too surprising that it can recapitulate some human abilities (such as understanding simple functions). Similarly, it is really surprising that GPT can make better predictions when it is given domain knowledge about the function than when it is not? Now, I certainly don't mean to denigrate the field of "GPT studies" here- carefully cataloguing and understanding the capabilities of this system is surely of great interest. But by the standards of making "new, relevant, impactful knowledge", I unfortunately don't think the paper meets this bar, as it doesn't seem as though there are any especially novel prompting techniques, analysis methods, or tasks introduced in the paper.

Methodological Weaknesses

4. From Pg 7. For the feature selection task, it is not fair to Yamada et. al’s method to claim that GPT4 is “more data efficient” when you did not actually test Yamada’s method on the smaller dataset! If you want to make this claim, you would need to control the dataset size for both methods, for example evaluate both GPT4 and Yamada on one dataset of size 10^4 and another on 10^2 (or whatever other numbers seem appropriate).

5. On this same task, it needs to be clarified exactly how the evaluation was done, simply saying “Accuracy is measured by an MLP trained using 10^4 samples” is not sufficient. Was the data used to select the features separate from the data used to train the MLP? Is the “accuracy” here the training accuracy or test accuracy? Why are there errorbars on the accuracy, but only a single set of features reported for each method? Does this mean that you only ran each feature selection method once, if so why?

6. When asking GPT to describe its decision rules for the CO2 task, how much credence should we place in this answer and why? After all, the propensity of GPT to hallucinate is so well-known that it is often written about in the NY times. For example, the "increasing rate of change" does not appear to even be present in the data, as the authors point out. But the more fundamental issue is that,even for the features that are actually present in the data, that GPT may possibly be mis-representing its own decision process (i.e. its predictions might actually depend on some other feature of the data than what it says). Is there reason to think that GPT can accurately introspect into its own decision process?

7.In the currency exchange rate task, the authors state “there is a risk of data contamination in this case”. Why didn’t they apply the same mitigation as in the CO2 task (i.e. add random noise), or seemingly any mitigation at all?


8.

>I cannot use normal GP directly since there are missing values (section c.4)

A GP model can seamlessly handle arbitrary patterns of missing data so I am not sure what is meant here

-there are many other examples of important technical in the experimental details which are not described- specifics are given in the Questions section

Presentational Weaknesses

9. Overall I found the writing quality and organization of the paper to be poor, here are specific examples:

>in contrast to conventional machine learning models (pg.2)

This claim is way too strong relative to the actual results- as far as I can tell, the only other model that was systematically evaluated was a small  MLP


>three precious metals (pg.4)

Precious metals are not mentioned in any preceding text

>venturing into diverse domains extending from physics (pg.1)

There is no data presented on a physics task

> In this work, we ask a more fundamental question of how capable these models are in terms of representing functions at a general level (pg.9)

What does it mean to represent a function at a “general level”, and how did the previously-mentioned papers fall short at this?

Section2 seems superfluous to me- as the formal Bayesian framework is never used thereafter, and the math is well-known enough that I don’t think this section adds anything to the understanding

While “Predicting missing parts” is listed as a task in section 3.1, I do not see any actual results for this task presented anywhere in the paper or appendix. (Many of the prompts however allude to missing parts). The closest thing seems to be the currency task, where the LLM predicts a kernel function which is used to impute missing values via GP inference, but this is not really a direction prediction of missing values.

Similarly, in Figure 2, there are appear to be “ghost tasks” of Ball trajectory prediction and Stock market prediction, which are not actually studied in the paper.

>(this is relevant for the ‘output dependency modeling‘ task).

There is no “output dependency modeling” task (ctrl-F “output dependency”)


10. I also found the language to be overly vague or confusing at times, here are a few examples:

>lays the groundwork for a deeper understanding of many recent studies (pg.9)

>Here, we target the distribution function rather than a simple regression function y = f (x, t) (pg.4)

>As static benchmarks are not ideal (pg.2)

**Questions:**

1. For the “Function class prediction”, it seems the results are only presented through subplot titles in Figure 3. Not only are these very small and hard to read, but the bigger issue is that they are just shown without any kind of comment or context. For example, there seems to be a very wide range of accuracies, with the accuracy for Gaussian being ~1, and for piecewise being ~0; why do you think some function classes might be so much harder for GPT4 than others? For the ones it gets wrong, what are its predictions (i.e. confusion matrix)? Does this pattern of accuracies match any human data? How do other models (such as an MLP or Gaussia process) perform at this classification task?

2. What was the decoding temperature (tau) for GPT4? Since the results are reported with error bars, I am assuming it was not deterministic-in this case, how was the value chosen?

3.
>It is worth emphasizing that prompting the model in a different way can change the results significantly (pg.5)

This is an interesting and important observation - did you systematically evaluate the effect of different prompting strategies on the model performance?

4. For the kernel prediction tasks, does GPT4 also predict the hyperparameter values for the kernels? It doesn’t seem like the prompts specifically ask for this, but maybe the model gives them anyway?  If not, then how did you get the predictions for the LLM kernel?

5. In table 4, why do the first two entries have 0 standard deviation?

6.For the CO2 and currency tasks, who exactly are the human experts?

7.Some of the promprts, for example in section c seem to have an interactive element, e.g.

>You can ask a few questions to me.

Does GPT4 take you up on this? If so, what questions does it ask and how do you answer them?

**Details Of Ethics Concerns:**

No concerns

---

> ### Author Response · Authors · 2023-11-22
> **Response to reviewer rQFr [1/4]**
>
> First of all, we would like to thank the reviewer for their precious time in reviewing our manuscript, and for their very thorough and constructive feedback. We address the interesting questions and issues raised by the reviewer below:
>
>
> > The paper is motivated by asking whether GPT4 has some knowledge of simple functions (an example is given of the trajectory of a cannonball), akin to human capabilities. In my opinion this is a solid research question, but I am concerned that many of the experiments have only a tenuous connection to this question. In particular, I am thinking of feature selection, income classification, and currency prediction tasks. These seem to be essentially classic ML tasks, without much connection to function learning per se (beyond the vacuous connection that any supervised learning problem can be cast as learning a function from inputs to outputs- but this is quite different from, e.g. the motivating example of the cannonball trajectory).
>
> **Response:** We thank the reviewer for the interesting comment. We believe that tasks like income classification are still highly correlated to our core research question. This is because any relationship y = f(x) can be regarded as a function, both for regression (like the one used to model ball trajectory) as well as classification (like the one in the income task). The only difference is whether the target y is continuous or discrete. (b) In terms of tasks like feature selection, it is useful in reflecting one’s understanding about function. For example, if the underlying function is almost linear, then knowing which features are important are equivalent to knowing which weights are large in the linear function. Similarly, in the currency prediction task we use kernels designed by GPT-4, which can also show how the model understands the relationship between data points. In our research, we find methods like feature selection often provide additional insights about how LLMs understand a particular function. We believe direct prediction just covers one facet of the full function modeling capabilities that we intend to capture.
>
>
> > Considering that the authors motivate their work by comparing to human abilities this is a bit jarring- is there any evidence that humans are especially good at these kinds of tasks? Note that for the synthetic function prediction tasks, there actually is fairly detailed psychological data available on peoples’ performance.
>
> **Response:** Although we motivated our study from human ability, we intentionally avoid any direct comparisons, as we focus on the difference between function modeling with and without domain knowledge for machine learning models. An apples to apples comparison against human subjects is beyond the scope of our work.
>
>
> > In my opinion, the authors do not give sufficient consideration to the issue of data leakage. To their credit, they do identify this as an issue and propose specific fixes. But my concern is that there is no evaluation of whether these fixes have actually done anything to mitigate the issue.( I do realize the authors are in a bit of a bind here- on the one hand, it is important to evaluate on real-world data, but on the other, details of the GPT4’s training data are unfortunately not publicly available. ) What I would propose is something like this (using the CO2 data for example): evaluate GPT4 on (a) the raw, unmodified data, (b) the data with random noise added, and ( c ) some number of synthetic datasets constructed with a similar latent structure of a combination of a small number of Gaussian kernels. If the performance on GPT4 on (a) is nearly perfect, but the performance on (b) is significantly degraded, and close to the performance on (c ), then I would be convinced that the data leakage mitigation procedure has been at least partially successful. I’m not saying the authors have to do exactly this demonstration- but there does need to be some evaluation of the effectiveness of the leakage mitigation techniques.
>
> **Response:** We agree with the reviewer that more thorough evaluation is needed to check the existence of data leakage, and we are thankful for your suggestion. In fact, in our experiment, we are already following a setup highly similar to (c), that we not only add random noises as in (b) but also re-normalized the data to be within [0,1] as well as perform feature renaming and merging. Despite this concern of data leakage, we believe that our dataset achieves precisely the effect that we would like to capture i.e., recognition based on something similar seen during training. Therefore, we consider this to still be a valid benchmark. Additionally, it is worth noting that the method suggested by the reviewer may not be ideal in our setup as it can only prove there is some leakage, while it is insufficient to prove no leakage (as the reviewer also noted).

---

> ### Author Response · Authors · 2023-11-22
> **Response to reviewer rQFr [2/4]**
>
> > This is a bit more subjective, but I can't help but feel that, even absent the issues described elsewhere in this review, that this paper doesn't make much of an original contribution beyond answering "how well can GPT do X" for yet another value of X. We already know that GPT can do a lot of things quite well, and considering it was trained on data generated by humans, it is not too surprising that it can recapitulate some human abilities (such as understanding simple functions). Similarly, it is really surprising that GPT can make better predictions when it is given domain knowledge about the function than when it is not? Now, I certainly don't mean to denigrate the field of "GPT studies" here- carefully cataloguing and understanding the capabilities of this system is surely of great interest. But by the standards of making "new, relevant, impactful knowledge", I unfortunately don't think the paper meets this bar, as it doesn't seem as though there are any especially novel prompting techniques, analysis methods, or tasks introduced in the paper.
>
> **Response:** It is true that there is already a plethora of work investigating ‘how GPT can do X’ on different tasks X. However, our work is substantially different in that (a) we try to investigate a more fundamental problem, i.e., which kinds of function modeling problems can be tackled by GPT, specifically when taking into account the capability to seamlessly integrate domain knowledge (see e.g. the synthetic task in our experiment). This provides a new perspective to understand these ‘can do X’ works; and (b) we are the first to study the function modeling ability of GPT from a Bayesian perspective, where we explicitly decouple the ability to recognize patterns from raw data and the ability to integrate with prior knowledge learned from the internet. Importantly, our work highlights the potential of LLM in modern data science, specifically its potential in cases where data is scarce or domain knowledge is important. Finally, we would like to point out that our evaluation framework is also new: we evaluate GPT’s understanding about function from different angles, using both direct methods (checking output accuracy) and indirect methods (feature selection/kernel design). The insights revealed by these different methods are complementary and cover a broad range of different applications.
>
>
> > From Pg 7. For the feature selection task, it is not fair to Yamada et. al’s method to claim that GPT4 is “more data efficient” when you did not actually test Yamada’s method on the smaller dataset! If you want to make this claim, you would need to control the dataset size for both methods, for example evaluate both GPT4 and Yamada on one dataset of size 10^4 and another on 10^2 (or whatever other numbers seem appropriate).
>
> **Response:** We thank the reviewer for pointing this out. We have now also provided the results for Yamada’s method in the small data regime. The results show that Yamada’s method is inferior to GPT-4 when using 10^3 samples, achieving an accuracy of ~80% compared to ~83% for our method. When considering 10^2 samples, which is the same case as our method, the method from Yamada et al. fails to perform sensible feature selection (it does not select features at all). We note that this is expected as Yamada’s method works by training a neural network, which is typically data hungry.
>
>
> > On this same task, it needs to be clarified exactly how the evaluation was done, simply saying “Accuracy is measured by an MLP trained using 10^4 samples” is not sufficient. Was the data used to select the features separate from the data used to train the MLP? Is the “accuracy” here the training accuracy or test accuracy? Why are there errorbars on the accuracy, but only a single set of features reported for each method? Does this mean that you only ran each feature selection method once, if so why?
>
> **Response:** The data used in feature selection and training classifier is the same. “Accuracy” in the table means test accuracy. The error bars in the table correspond to different runs of ‘feature selection + subsequent classifier training based on the selected features’. In this process, the feature selection results are often stable, however the performance of the trained classifiers can vary across different runs due to e.g. different initialization/optimization dynamics. This is why there are error bars even if the same set of features are selected.

---

> ### Author Response · Authors · 2023-11-22
> **Response to reviewer rQFr [3/4]**
>
> > When asking GPT to describe its decision rules for the CO2 task, how much credence should we place in this answer and why? After all, the propensity of GPT to hallucinate is so well-known that it is often written about in the NY times. For example, the "increasing rate of change" does not appear to even be present in the data, as the authors point out. But the more fundamental issue is that,even for the features that are actually present in the data, that GPT may possibly be mis-representing its own decision process (i.e. its predictions might actually depend on some other feature of the data than what it says). Is there reason to think that GPT can accurately introspect into its own decision process?
>
> **Response:** That’s a great question. We completely agree that the model can misrepresent the actual decision rule that it is using. Therefore, this evaluation just validates that GPT-4 has knowledge about the domain, rather than validating what GPT-4 is doing under the hood (Huang et al. 2023 attempts to do a systematic comparison against classical interpretability techniques). We updated the paper to make this clear.
>
> *Huang, S., Mamidanna, S., Jangam, S., Zhou, Y. and Gilpin, L.H., 2023. Can Large Language Models Explain Themselves? A Study of LLM-Generated Self-Explanations. arXiv preprint arXiv:2310.11207.*
>
>
> > In the currency exchange rate task, the authors state “there is a risk of data contamination in this case”. Why didn’t they apply the same mitigation as in the CO2 task (i.e. add random noise), or seemingly any mitigation at all?
>
> **Response:** We appreciate your attention to the issue of data contamination in our currency exchange rate task. To mitigate this, we implemented a two-pronged approach similar to our CO2 task methodology. First, we introduced random noise into the raw values to obscure direct data correlations. Second, we removed specific identifiers such as currency and metal names that could directly link the data to our training set (we mistakenly included the different version of the prompt, which was not used in the experiment in Appendix c.4, which we have fixed in the recent version – we apologize for the confusion that it created). This dual approach ensures the integrity of our data while maintaining its usefulness for training. We acknowledge that this was not explicitly stated in our initial submission, and we will clarify this methodology in the revised manuscript.
>
>
> > I cannot use normal GP directly since there are missing values (section c.4). A GP model can seamlessly handle arbitrary patterns of missing data so I am not sure what is meant here
>
> **Response:** We are thankful to the reviewer for highlighting this, and apologize for the confusion. We highlighted the deficiency of normal GP in this prompt as we would also like to also see whether GPT-4 can come up with solutions beyond default missing data impulation techniques. For example, a good solution will consider the correlation between different dimensions rather than treating them separately during missing data imputation. The corresponding result was not included in the final manuscript for conciseness.
>
>
> > Organization and writing
>
> **Response:** We are thankful to the reviewer for the detailed comments regarding the writing and organization of the paper. We addressed all the organizational and writing related issues mentioned by the reviewer, and did another pass over the entire paper, attempting to improve clarity. All recent changes are highlighted in RED in order to make them clear.
>
>
> > Section2 seems superfluous to me- as the formal Bayesian framework is never used thereafter, and the math is well-known enough that I don’t think this section adds anything to the understanding
>
> **Response:** We attempted to lay out the Bayesian framework in terms of learning a useful prior from pretraining, and tie it back in the end. We think that it still adds the right context to our study, laying out the basic idea of viewing LLM pretraining as a Bayesian prior. In fact, in the experiments, we strictly follow this framework by discerning the ability of LLM to model the likelihood term (when only raw data is provided) and its ability to model the full posterior (when both data + context information are provided).

---

> ### Author Response · Authors · 2023-11-22
> **Response to reviewer rQFr [4/4]**
>
> ***Responses to questions raised by the reviewer***
>
> > For the “Function class prediction”, it seems the results are only presented through subplot titles in Figure 3. Not only are these very small and hard to read, but the bigger issue is that they are just shown without any kind of comment or context. For example, there seems to be a very wide range of accuracies, with the accuracy for Gaussian being ~1, and for piecewise being ~0; why do you think some function classes might be so much harder for GPT4 than others? For the ones it gets wrong, what are its predictions (i.e. confusion matrix)? Does this pattern of accuracies match any human data? How do other models (such as an MLP or Gaussia process) perform at this classification task?
>
> **Response:** We are thankful to the reviewer for highlighting this. We agree that the intelligibility of the results can be significantly improved. Although the rebuttal period itself was insufficient to complete the task, we intend to include a more thorough discussion on the results (including confusion matrices), as well as comparison against other ML baselines in the next version of the paper.
>
>
> > What was the decoding temperature (tau) for GPT4? Since the results are reported with error bars, I am assuming it was not deterministic-in this case, how was the value chosen?
>
> **Response:** Although we chose a temperature of 0, it is not deterministic due to problems with GPT-4 determinism as popularly known (they might have fixed it as claimed in the recent dev conference).
>
>
> > It is worth emphasizing that prompting the model in a different way can change the results significantly (pg.5). This is an interesting and important observation - did you systematically evaluate the effect of different prompting strategies on the model performance?
>
> **Response:** We agree that it is an important point, just like in any LLM paper. However, it is a particularly hard one to evaluate. In this paper, we attempted to identify the utility that can be extracted by an average practitioner by simply prompting the model using simple language. Therefore, optimizing the performance of the prompt to exact the maximum performance out of the model was beyond the scope of our work. We mainly mentioned this as a reminder and a caveat that the reported numbers are a lower-bound on performance.
>
>
> > For the kernel prediction tasks, does GPT4 also predict the hyperparameter values for the kernels? It doesn’t seem like the prompts specifically ask for this, but maybe the model gives them anyway? If not, then how did you get the predictions for the LLM kernel?
>
> **Response:** Thank you for your insightful query regarding the kernel prediction tasks. We clarify that except for the case where we explicitly ask for specific hyperparameter values, GPT-4 will not output them for us. We also highlight that throughout our experiments, we have not asked about hyperparameter recommendation as we attempt to investigate the role of GPT-4 as a data-scientist, who can predict the right model, but not the right hyperparameters for these models. We agree that investigating whether the hyperparameters suggested by GPT-4 are useful will be an interesting topic to explore in the future.
>
>
> > In table 4, why do the first two entries have 0 standard deviation?
>
> **Response:** This is because in these two cases, the standard deviation in indeed very low as compared to other cases.
>
> > For the CO2 and currency tasks, who exactly are the human experts?
>
> **Response:** Thank you for validating this. We took the best practices from state-of-the-art research papers. Hence, they form the experts in our case. We made this more explicit in the paper.
>
> > Some of the prompts, for example in section c seem to have an interactive element, e.g. You can ask a few questions to me. Does GPT4 take you up on this? If so, what questions does it ask and how do you answer them?
>
> **Response:**  We are thankful to the reviewer for noticing these details, which would like to clarify below. In most cases, GPT-4 asked for clarification about tasks, such as the task goal and the meaning of features (e.g. ‘what do you mean by net capital gain’ in the income task). We provide this necessary information to GPT-4 in such cases. For other cases which involve queries about the sensitive information that can be used to infer the dataset (e.g. ‘which observatory are these data collected from’), we refuse to answer to avoid potential data leakage.

---

> ### Comment · Reviewer_rQFr · 2023-11-22
> **Response to rebuttal**
>
> Thank you for your thorough response to my questions and comments! I believe that most of the technical and presentational issues that I raised have been addressed satisfactorily. However, after having read your response, I still stand by my "conceptual" points 1-3. I will elaborate below:
>
> Point 1
>
> >any relationship y = f(x) can be regarded as a function, both for regression (like the one used to model ball trajectory) as well as classification (like the one in the income task). The only difference is whether the target y is continuous or discrete
>
> I agree with this observation, and I apologize if I was not clear in my initial comment. My concern is not that the income task (e.g.) is not a function (which I agree it is), but rather that is it not particularly *simple* (at least by human standards), considering that the domain has 13 dimensions, compared to 1 for the motivating cannonball task. Now, you clarify below that your intention is not to directly compare with human data: ok, fair enough. In that case, I think the motivation and references to human abilities,intuitive physics,etc. are a bit distracting and should be removed. But more importantly, it is then very unclear what exactly constitutes the class of "admissible" functions that you study in the paper, and why you choose to study this class in particular. If anything, I am a bit more confused now, because I originally thought that "human abilities" were being used to define the class of functions of interest; this is a reasonably well-defined and interesting class. But now I am confused about what the class of functions even is. In the introduction, you state "we ask...how capable these models are in terms of representing general functions", but I don't think that "general" is the right descriptor. To illustrate this point,one could imagine a task where a LLM is given a very large composite number and asked to output its prime factors; this is a "general" function in the mathematical sense, although I would guess that you would consider it to fall out of the scope of functions you are interested in for the purposes of the current study (correct me if I am wrong). So there is some restricted class of functions that you are considering, it is not completely "general", and it is unclear to me what the class is and why you chose it in particular.
>
> Point 2
>
> Thank you for clarifying your setup; i do agree that it is similar to my suggestions (b) and (c). But, as I explained in the following sentence, the key part of my proposal is to *compare* the performance of (a) with the performance of (b) and (c), which AFAICT has not been done. But as I said, I am not wedded to this method in particular- it is just an example of the sort of thing I would find compelling. I admittedly am not an expert in the particular domain of data leakage with LLMs- one of the other reviewers appears to have more detailed knowledge here. But without any quantitative/systematic demonstration of the efficacy of your leakage mitigation techniques, I am afraid I cannot agree that the data is "still...a valid benchmark".
>
> Point 3
>
> Thank you for clarifying the scope of your contributions. I agree that your points (a) and (b) are potentially compelling directions, however I simply don't think that the current paper has enough to say about them in order for them to be considered the key contributions. Regarding (a), I actually found the domain-knowledge experiments to be the least compelling parts of the paper; as I said elsewhere, it doesn't seem surprising at all that GPT would perform better when it is given more information about its input. If there are situations where it uses the domain knowledge especially effectively or especially ineffectively, for example, that could be quite interesting. Regarding (b), I still feel that the Bayesian perspective is only really used in a metaphorical sense. Without any kind of quantiative estimation of, e.g., the prior strength or form of the likelihood function, I am afraid I cannot consider this a substantive contribution of the paper.
>
> Again, thank you for your thorough attention to my comments, and I do believe that aspects of the paper have been significantly improved. But in light of the remaining conceptual issues, I am opting to keep my score the same.

---

> > ### Author Response · Authors · 2023-11-22
> > **Response to reviewer rQFr's response [1/2]**
> >
> > We are very thankful to the reviewer for responding to our rebuttal on such short notice. We respond to the reviewer's points below.
> >
> > > Point 1
> >
> > > any relationship y = f(x) can be regarded as a function, both for regression (like the one used to model ball trajectory) as well as classification (like the one in the income task). The only difference is whether the target y is continuous or discrete
> >
> > > I agree with this observation, and I apologize if I was not clear in my initial comment. My concern is not that the income task (e.g.) is not a function (which I agree it is), but rather that is it not particularly simple (at least by human standards), considering that the domain has 13 dimensions, compared to 1 for the motivating cannonball task. Now, you clarify below that your intention is not to directly compare with human data: ok, fair enough. In that case, I think the motivation and references to human abilities,intuitive physics,etc. are a bit distracting and should be removed. But more importantly, it is then very unclear what exactly constitutes the class of "admissible" functions that you study in the paper, and why you choose to study this class in particular. If anything, I am a bit more confused now, because I originally thought that "human abilities" were being used to define the class of functions of interest; this is a reasonably well-defined and interesting class. But now I am confused about what the class of functions even is. In the introduction, you state "we ask...how capable these models are in terms of representing general functions", but I don't think that "general" is the right descriptor. To illustrate this point,one could imagine a task where a LLM is given a very large composite number and asked to output its prime factors; this is a "general" function in the mathematical sense, although I would guess that you would consider it to fall out of the scope of functions you are interested in for the purposes of the current study (correct me if I am wrong). So there is some restricted class of functions that you are considering, it is not completely "general", and it is unclear to me what the class is and why you chose it in particular.
> >
> > **Response:**
> > Thanks a lot for your clarification. We are now in a better position to understand your concerns. Let us discuss them from two aspects. (a) the function in the income task, though its high input dimensionality, may still be intuitively modeled by humans using common sense. For example, occupation and education are well-known correlated to income, so that an educated individual can intuitively make reasonable predictions using these features without knowing the exact form of the function. In this sense, it is still very much similar to the ball trajectory example; (b) we apologize for missing a clear definition of the function of interests in this work. To clarify, the functions we study in this work can be very broad, that it can be any function in the real-world, so long as domain knowledge is helpful in understanding it.  The reviewer is correct in pointing out that functions such as the prime factorization falls beyond the scope of the considered functions since we cannot make intuitive predictions about them i.e., the only way to solve them is to exactly solve the problem. We would also like to clarify that the ball trajectory task was used just as a motivating example to highlight that ‘domain knowledge is helpful when modeling the underlying function’ rather than ‘this task can be easily dealt with by humans’. Finally, we clarify that when talking about ‘whether LLM can understand function intuitively like humans’, we are referring to its *thinking patterns* being similar to humans i.e. whether it can model function without knowing its explicit analytic form and whether it can make use of domain knowledge to form an implicit understanding such that it can make effective predictions without understanding the underlying processes well, rather than comparing this ability directly with humans.

---

> ### Author Response · Authors · 2023-11-22
> **Response to reviewer rQFr's response [2/2]**
>
> > Point 2
>
> > Thank you for clarifying your setup; i do agree that it is similar to my suggestions (b) and (c). But, as I explained in the following sentence, the key part of my proposal is to compare the performance of (a) with the performance of (b) and (c), which AFAICT has not been done. But as I said, I am not wedded to this method in particular- it is just an example of the sort of thing I would find compelling. I admittedly am not an expert in the particular domain of data leakage with LLMs- one of the other reviewers appears to have more detailed knowledge here. But without any quantitative/systematic demonstration of the efficacy of your leakage mitigation techniques, I am afraid I cannot agree that the data is "still...a valid benchmark".
>
> **Response:**
> We are glad that we have a consensus that our setup is indeed similar to (b)(c) — this means we actually share the same ideas in avoiding data leakage. In terms of a comparison between (a) and (b)(c), we completely agree that such explicit comparisons are highly useful. However, an implicit comparison as done in the current paper can also tell a sensible story. Specifically, let t be the ground truth for this task. Now we already know the gap |(c) - t| is small. Since (c) is a lower bound case of (a), we know the gap |(a) - t| < |(c) - t|, being also small. This immediately yields that |(c) - (a)| is also small due to triangular inequality. As discussed in our first response, if (a)(c) are similar, it can hardly tell whether there is dataset leakage or not. On the other hand, we do have results for comparing (a) and (c). We intentionally choose to not report these results since it is orthogonal to the main story and helps little in proving the non-existence of data leakage. We will add the numerical comparison between (a)(c) again as well as including the discussion above in the appendix. Finally, we would like to thank you again for raising our attention to potential data leakage as well as the call on a systematic/quantitative demonstration, which is constructive.
>
> > Point 3
>
> > Thank you for clarifying the scope of your contributions. I agree that your points (a) and (b) are potentially compelling directions, however I simply don't think that the current paper has enough to say about them in order for them to be considered the key contributions. Regarding (a), I actually found the domain-knowledge experiments to be the least compelling parts of the paper; as I said elsewhere, it doesn't seem surprising at all that GPT would perform better when it is given more information about its input. If there are situations where it uses the domain knowledge especially effectively or especially ineffectively, for example, that could be quite interesting. Regarding (b), I still feel that the Bayesian perspective is only really used in a metaphorical sense. Without any kind of quantiative estimation of, e.g., the prior strength or form of the likelihood function, I am afraid I cannot consider this a substantive contribution of the paper.
>
> **Response:**
> Thank you for your constructive critique. Regarding your comments on (a), we again agree that we already know GPT can do a good job in a multitude of tasks. However, a work showing LLM’s potential in a new task (data science) may still be valuable to the community, especially that we reveal its strengths in cases with limited data and cases where domain knowledge is important. Our idea to treat LLMs as pre-trained universal function approximators for real-world functions is also new in the community to our knowledge. Regarding your comments on (b), we agree that a qualitative assessment to the impact of prior will be very attractive and helpful. But this might go well beyond our original research goal which focus on evaluating LLM’s *intuitive* understanding about functions. Importantly, despite the absence of an analytic likelihood function, we are still able to reveal LLMs' deficiency in function modeling with our framework, that it is indeed weak in modeling the likelihood function (see e.g. the case where there is only raw data presented), in comparison to its strong ability in integrating prior knowledge with data patterns. This discovery is also new to the community.
>
>
> ***Final remarks:***
> We would like to express our appreciation for your time and effort in the review and discussion, particularly the constructive criticism. The comments from the reviewer helped us in improving the quality of our draft.

---

### Meta-Review · Area_Chair_udtf · 2023-12-06

**Metareview:**

The paper studies an interesting problem -- the domains are interesting and the experiments compelling, interesting evaluation structure, the application to data science is cool. However, the overall contribution of the work is a bit narrow (only evaluating a relatively small / bespoke dataset on a single pre-trained model). Evaluating on a wider range of models (pretrained or from scratch transformers), generalizing the dataset to be larger, or doing further quantiative formalization and analyses of how the prior and likelihood info are combined would push this over the line.

**Justification For Why Not Higher Score:**

too narrow, only evaluating a relatively small / bespoke dataset on a single pre-trained model

**Justification For Why Not Lower Score:**

NA

---

### Decision · Program_Chairs · 2024-01-16

Reject